# Transcriptome and epigenome diversity and plasticity of muscle stem cells following transplantation

Brendan Evano[1,2]*, Diljeet Gill[3], Irene Hernando-Herraez[3], Glenda Comai[1,2], Thomas M. Stubbs[3¤], Pierre-Henri Commere[4], Wolf Reik[3‡]*, Shahragim Tajbakhsh[1,2‡]*

1 Stem Cells & Development, Department of Developmental & Stem Cell Biology, Institut Pasteur, 25 rue du Dr. Roux, Paris, France, 2 CNRS UMR 3738, Institut Pasteur, Paris, France, 3 Epigenetics Programme, Babraham Institute, Cambridge, United Kingdom, 4 Cytometry and Biomarkers, Center for Technological Resources and Research, Institut Pasteur, 28 rue du Dr. Roux, Paris, France

☯ These authors contributed equally to this work.
¤ Current address: Chronomics Limited, 1 St James Court, Norwich.
‡ WR and ST also contributed equally to this work.
* brendan.evano@pasteur.fr (BE); wolf.reik@babraham.ac.uk (WR); shahragim.tajbakhsh@pasteur.fr (ST)

**Data Availability Statement:** Sequencing data will be available in GEO with the accession GSE154813 after acceptance.

## Abstract

Adult skeletal muscles are maintained during homeostasis and regenerated upon injury by muscle stem cells (MuSCs). A heterogeneity in self-renewal, differentiation and regeneration properties has been reported for MuSCs based on their anatomical location. Although MuSCs derived from extraocular muscles (EOM) have a higher regenerative capacity than those derived from limb muscles, the molecular determinants that govern these differences remain undefined. Here we show that EOM and limb MuSCs have distinct DNA methylation signatures associated with enhancers of location-specific genes, and that the EOM transcriptome is reprogrammed following transplantation into a limb muscle environment. Notably, EOM MuSCs expressed host-site specific positional *Hox* codes after engraftment and self-renewal within the host muscle. However, about 10% of EOM-specific genes showed engraftment-resistant expression, pointing to cell-intrinsic molecular determinants of the higher engraftment potential of EOM MuSCs. Our results underscore the molecular diversity of distinct MuSC populations and molecularly define their plasticity in response to microenvironmental cues. These findings provide insights into strategies designed to improve the functional capacity of MuSCs in the context of regenerative medicine.

## Author summary

Adult skeletal muscles are regenerated upon injury by muscle stem cells (MuSCs). A heterogeneity in expression of key myogenic regulators and regeneration properties has been reported for MuSCs based on their anatomical location. Although MuSCs derived from extraocular muscles (EOM) have a higher regenerative capacity than those derived from limb muscles, the molecular determinants that govern these differences remain undefined.

**Funding:** This project was supported by grants from Institut Pasteur, Agence Nationale de la Recherche (Laboratoire d'Excellence Revive, Investissement d'Avenir; ANR-10-LABX-73), Association Française contre les Myopathies (21857), CNRS, the European Research Council (Advanced Research Grant 332893), and the Biotechnology and Biological Sciences Research Council (BBSRC, CBBS/E/B/000C0425). I.H.-H. was supported by a Marie Sklodowska-Curie Individual Fellowship (751439). The funders had no role in study design, data collection and analysis, decision to publish, or preparation of the manuscript.

**Competing interests:** I have read the journal's policy and the authors of this manuscript have the following competing interests: W.R. is a consultant and shareholder of Cambridge Epigenetix. T.S. is CEO and shareholder of Chronomics. All other authors declare no competing interests.

Here we show that EOM and limb MuSCs have distinct transcriptome and DNA methylation signatures, and that the EOM transcriptome is reprogrammed following transplantation into a limb muscle environment. Notably, EOM MuSCs adopted host-site specific positional *Hox* codes after engraftment within the host muscle. However, about 10% of EOM-specific genes were resistant to alterations following heterotopic engraftment, pointing to molecular determinants of the high engraftment potential of EOM MuSCs. Our results underscore the molecular diversity of distinct MuSC populations and molecularly define their plasticity in response to microenvironmental cues. These findings provide insights into strategies designed to improve the functional capacity of MuSCs in the context of regenerative medicine.

## Introduction

Skeletal muscles are essential for physiological functions such as locomotion, breathing, and metabolism, and they represent up to 40% of the human body mass. Tissue-specific muscle stem (satellite) cells (MuSCs) ensure skeletal muscle homeostasis and regeneration [1–3]. MuSCs have been implicated in the etiology of some muscular dystrophies [4,5] as well as age-associated impaired muscle regeneration [6–11] leading to an incapacitating decrease of muscle mass and strength [12–15]. Stem-cell therapies have proven to be challenging for muscular dystrophies, as they require delivering enough functional MuSCs to the right muscle groups and *ex vivo* amplification of healthy MuSCs results in a major decline in their regenerative capacity and stemness properties [16].

Most of our knowledge on MuSC biology arises from the study of limb muscles, whereas MuSCs from other muscle groups remain less well characterized. Extraocular muscles (EOMs) are responsible for eye movements, with the basic pattern of 6 muscles conserved in most vertebrate species [17]. Limb muscles derive from somitic mesoderm and they rely in part on *Pax3* expression and function [18–20]. In contrast, EOMs derive from cranial mesoderm and rely initially on *Mesp1* and *Pitx2* for their emergence, yet unlike the majority of other cranial-derived muscles, their founder stem cell population [21,22] arises independently of *Tbx1* function [17,23–29]. After the distinct specification of cranial and trunk progenitors, the core myogenic regulatory factors Myf5, Mrf4, Myod, and Myogenin regulate myogenic cell commitment and differentiation [30]. In adult homeostatic muscles, MuSCs are mostly quiescent. They are activated upon muscle injury, proliferate and differentiate to contribute to new muscle fibers, or self-renew to reconstitute the stem cell pool [31–34]. This process is accompanied by a temporal expression of cell fate markers, such as the transcription factors Pax7 (stem), Myod (commitment) and Myogenin (differentiation). Several reports indicate that EOMs are functionally different from their hindlimb counterparts since they are preferentially spared in ageing and several muscular dystrophies [35–38]. Interestingly, adult EOM-derived MuSCs cells have superior *in vitro* proliferation, self-renewal and differentiation capacities, as well as a superior *in vivo* engraftment potential, compared to limb-derived MuSCs [39]. These properties were maintained by EOM-derived MuSCs from dystrophic or aged mice [39]. However, the unique functional properties of adult EOM MuSCs remain undefined at the molecular level, and it is unclear if their specificity is instructed by cell-intrinsic factors or through interactions with their microenvironment (niche).

DNA methylation is a critical epigenetic mechanism involved in establishing and maintaining cellular identity. Dynamic changes in expression of DNA (de)methylation enzymes were reported in MuSCs during muscle regeneration [40–42], indicating a potential increase of

DNA methylation from quiescent to activated MuSCs. Several studies investigated the DNA methylation signatures of proliferating and differentiating cultures of MuSCs [43–45] and reported some changes in DNA methylation patterns during late myogenic differentiation. Further, aged MuSCs show increased DNA methylation heterogeneity at promoters, associated with a degradation of coherent transcriptional networks [46]. However, whether quiescent MuSCs from homeostatic muscles at different anatomical locations display similar or different DNA methylation patterns remains unknown.

Here we performed parallel DNA methylation and transcriptome sequencing to character-ize the specific identity of adult mouse EOM and limb MuSCs at the molecular level. We used heterotopic transplantation of EOM MuSCs into limb muscles to challenge their fate and assess their plasticity. We show that their specific identity is mostly niche-driven as they adopt a limb-like molecular signature. Nevertheless, we also identify EOM-specific genes that resist reprogramming by the microenvironment, indicating potential candidates to manipulate in limb-derived MuSCs for improving their regenerative capacity in the context of cellular therapies.

## Results

To investigate the molecular differences between cranial- and limb-derived muscles, we iso-lated MuSCs from EOM and *Tibialis anterior* (TA) by FACS (Fluorescence-Activated Cell Sorting) from 10 adult *Tg*:*Pax7-nGFP* [17] mice and processed them for RNA-sequencing and BS-sequencing as in [47,48] (Fig 1A). EOM and TA MuSCs showed similar FACS profiles (S1A Fig). TA MuSCs showed a slightly higher GFP fluorescence intensity than EOM MuSCs, which correlates with the difference in *Pax7* expression as observed through RNA-sequencing (Fig 1D). Unsupervised Principal Component Analysis (PCA) on the transcriptomes revealed a clear discrimination between EOM and TA MuSC samples, indicating location-specific tran-scriptional identities (Fig 1B). To further investigate these transcriptional signatures, we per-formed differential expression analysis and found 261 genes significantly upregulated in EOM stem cells (EOM Differentially Expressed Genes, DEGs) and 339 genes significantly upregu-lated in TA MuSCs (TA DEGs) (DE-seq, p <0.05, FC >2) (Fig 1C). The muscle stem cell marker *Pax7* and the myogenic factors *Myf5* and *Myod* showed slightly higher expression lev-els in TA compared to EOM MuSCs, while the transcription factor *Pitx2* showed higher expression levels in EOM MuSCs (Fig 1D). As reported [49], *Pax3* was expressed in TA MuSCs and not detected in EOM MuSCs. Other genes specifically up-regulated in TA MuSCs included the developmental factors *Lbx1 and Tbx1*. Lbx1 is a homeobox transcription factor required for the migration of myogenic progenitor cells to the limbs, and Tbx1 is a T-box containing transcription factor necessary for the development of craniofacial muscles [50–52]. Examples of EOM up-regulated genes included the homeobox transcription factors *Lmx1a* and *Alx4* [30,53,54] as well as *Mobp* (involved in myelination [55]) (Fig 1D). Further, genes of the *Hox* gene family were upregulated in TA samples (Fig 1E), consistent with their anteroposterior expression pattern in vertebrates [56]. Overall, 13 *Hox* genes were upregulated, representing 64% of the *HoxA* cluster and 75% of the *HoxC* cluster. Notably, these *Hox* genes are expressed more posteriorly along the body axis [57]. Overall, differentially expressed genes were enriched for developmental processes suggesting that location-specific patterns reflect their different origins during embryogenesis (S1B Fig).

Overall, global DNA methylation levels were similar between EOM and TA MuSCs, with those from the EOM showing slightly higher levels of DNA methylation relative to TA MuSCs. On average genome-wide DNA methylation levels were around 50%. Gene bodies were meth-ylated at similar levels to the genome-wide average (50%), whereas, promoters, enhancers and

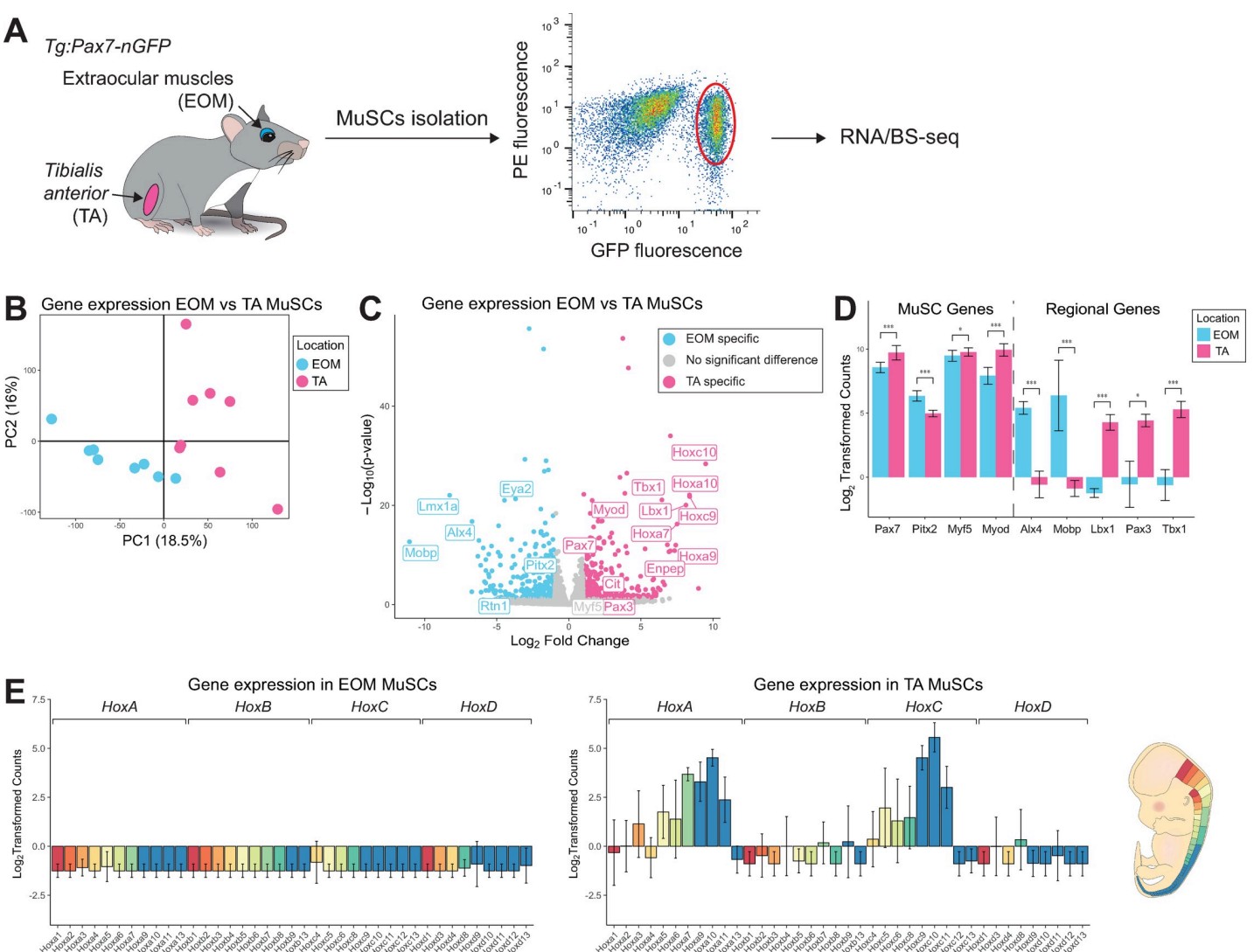

**Fig 1. Head and limb-derived MuSCs display specific transcriptome signatures.** (A) Experimental scheme. Head and limb MuSCs were isolated by FACS from EOM and TA muscles respectively, from *Tg:Pax7-nGFP* adult mice and processed through RNA- and bisulfite-sequencing. N = 10 mice, n = 500 cells/mouse (S1 Table). (B) PCA analysis of EOM and TA MuSC transcriptomes. The first principal component separates samples based on their anatomical location. (C) Volcano plot showing the results of differential expression analysis of EOM and TA MuSCs. Genes that demonstrated a fold change greater than 2 and a p-value less than 0.05 according to DESeq2 were classified as differentially expressed and colour-coded by their tissue specificity. (D) Selected markers and differentially expressed genes between EOM and TA MuSCs. Error bars represent the standard deviation of the mean. p-values were determined by DESeq2. *** p < 0.001, ** p < 0.01, * p < 0.05. (E) Gene expression analysis throughout the *HoxA*, *HoxB*, *HoxC* and *HoxD* clusters in EOM (left) and TA (middle) MuSCs. Genes were colour-coded according to their antero-posterior expression domain in mouse at embryonic day 12.5 (right, adapted from [57]).

CpG islands were hypomethylated (25%, 30% and 5% respectively) and repeat elements were hypermethylated (85%) (Fig 2A). We restricted our more detailed analysis to enhancers (regions marked by H3K27Ac in MuSCs that did not overlap promoters) [58] and promoters (-2000bp to 500bp of the TSS of Ensembl genes) and calculated DNA methylation levels over each genomic element. PCA analysis for promoter regions showed no clear differences between EOM and TA MuSCs (Fig 2B). In contrast, a clear separation by anatomical location was observed when considering enhancer regions (Fig 2C). This clustering was also observed when restricting the analysis to 544 enhancers associated with EOM and TA DEGs (enhancer regions within 1 Mb to the closest transcription start site, S2A Fig). Furthermore, enhancers

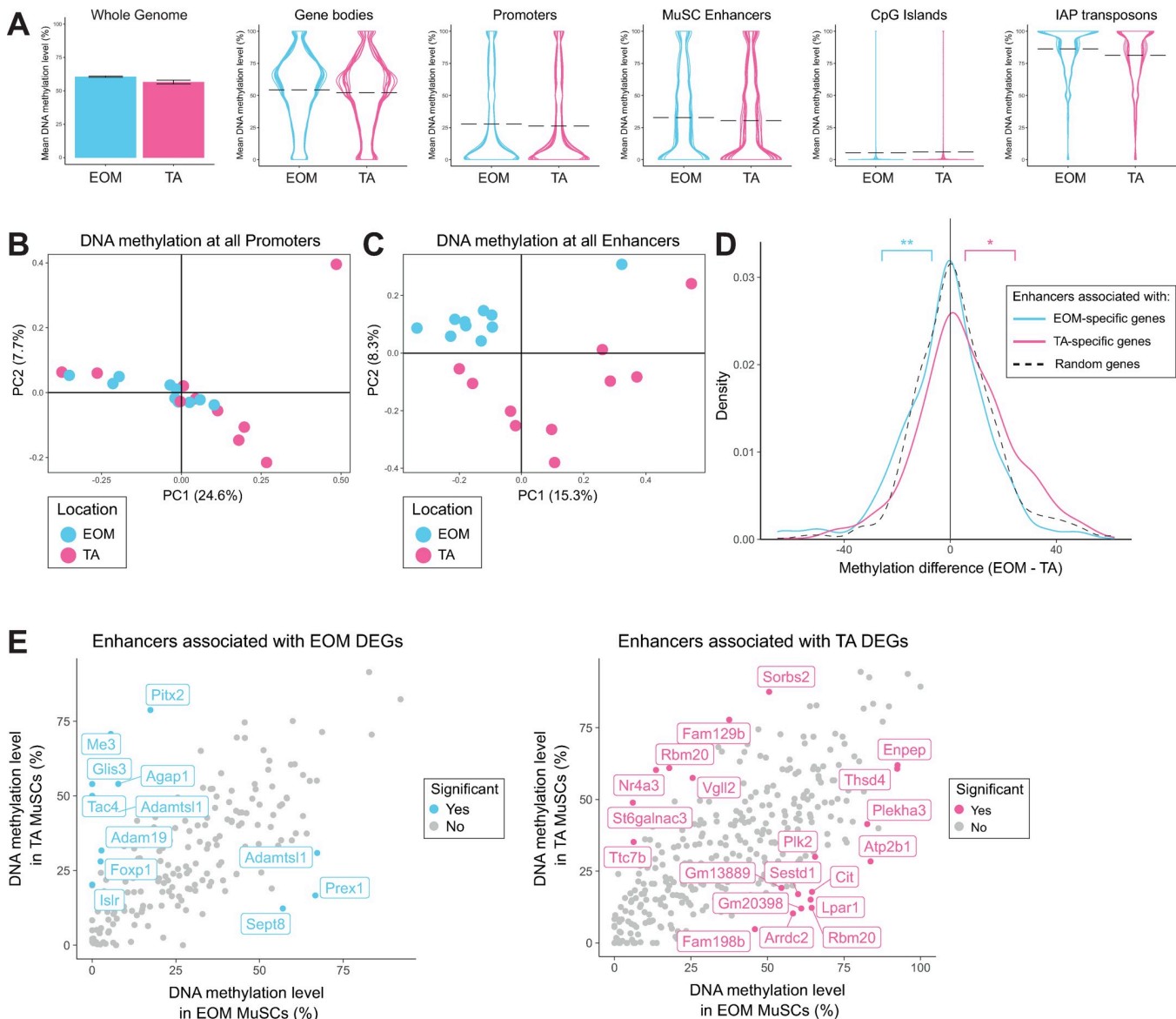

**Fig 2. Head and limb-derived MuSCs display specific DNA methylation patterns at enhancers.** (A) DNA methylation levels of the whole genome, promoters, enhancers, gene bodies, CpG islands and IAP repeat elements in EOM and TA MuSCs. The mean DNA methylation level across the whole genome is represented as a bar chart with error bars representing the standard deviation between samples. The mean DNA methylation levels of individual genomic elements are represented as violin plots to highlight their distribution. Each line in the violin plots represents a different sample, mean values are indicated by dashed lines. Promoters were defined as -2000bp to 500bp of the TSS of Ensembl genes. H3K27ac peaks were called using macs2 on H3K27ac ChIP-seq data obtained from [58]. Enhancers were defined as H3K27ac peaks that did not overlap promoters. Enhancers were linked to genes based on proximity. Overall DNA methylation was similar between the two locations, though EOM MuSCs had slightly higher levels of DNA methylation relative to TA MuSCs. (B) PCA analysis of DNA methylation at promoters fails to separate EOM and TA samples based on their anatomical location. (C) PCA analysis of DNA methylation at enhancers separates EOM and TA samples based on their anatomical location. (D) Density plots showing methylation differences between EOM and TA MuSCs at enhancers associated with location-specific genes (enhancer regions within 1 Mb to the closest transcription start site of a DEG). Enhancers associated with EOM genes (n = 198) were significantly hypomethylated in EOM relative to TA MuSCs when compared to a random subset of enhancers (n = 350), while enhancers associated with TA genes (n = 346) were significantly hypermethylated in the same comparison. ** p < 0.01, * p < 0.05 by Welch's t test. (E) Scatter plots comparing the mean DNA methylation levels of enhancers associated with location DEGs in EOM and TA MuSCs. Enhancers were colour-coded as significant if they were also found to be differentially methylated by a rolling Z score approach (p < 0.05) when comparing the methylation levels of all enhancers in EOM and TA MuSCs (S2B Fig). The significant enhancers have been labelled with their associated gene. Only a subset of enhancers associated with location DEGs were differentially methylated. More enhancers associated with EOM DEGs were hypermethylated in TA than in EOM MuSCs. Likewise, more enhancers associated with TA DEGs were hypermethylated in EOM than in TA MuSCs.

associated with genes specifically upregulated in TA MuSCs were less methylated in TA MuSCs compared to EOM MuSCs. Similarly, enhancers associated with genes upregulated in EOM MuSCs were less methylated in EOM MuSCs compared to TA MuSCs (Fig 2D). To examine which enhancers were responsible for shifts in the distributions, we determined which enhancers were significantly differentially methylated using a rolling Z-score. Not all of the enhancers associated with EOM or TA DEGs were classified as differentially methylated, suggesting only a subset of these genes was regulated by enhancer methylation (Fig 2E). In addition, in some cases, multiple enhancers showing opposite methylation differences were associated with one DEG (such as *Adamtsl1*, with one enhancer hypermethylated in EOM MuSCs and one hypermethylated in TA MuSCs)(Fig 2E). This may suggest enhancer switching, which could be responsible for the expression difference if one of the enhancers was stronger than the other. Alternatively, only one of the enhancers may be responsible for controlling expression, which may be the one that is closer to the transcription start site. We also identified 29 Differentially Methylated Regions (DMRs) with more than 5 consecutive CpG methylation sites and 10% methylation differences. The majority of the DMRs overlapped enhancer or promoter regions such as the DMR overlapping the *Tbx1* promoter which is hypomethylated in TA MuSCs (Mean methylation levels in TA: 13%; mean methylation levels in EOM: 80%). These results suggest that DNA methylation patterns at regulatory regions contribute to the location-specific transcriptional profile of MuSCs.

To investigate MuSC plasticity and the influence of the cellular microenvironment on the observed location-specific signatures, we performed heterotopic transplantations of EOM MuSCs into TA muscles. Specifically, EOM MuSCs from *Tg:Pax7-nGFP* mice were transplanted into pre-injured TA muscles of immunodeficient $Rag2^{-/-};\gamma C^{-/-}$ [59] mice. As a control, and for each donor mouse, TA MuSCs were transplanted into the same recipient mouse, in the contralateral pre-injured TA muscle. After 28 days when muscle regeneration was complete and self-renewal of MuSCs could be evaluated, engrafted EOM and TA MuSCs were re-isolated by FACS based on GFP positivity (post-graft samples). Additionally, a fraction of the EOM and TA MuSCs was kept as a control before grafting (pre-graft samples). Pre-graft and post-graft EOM and TA MuSCs were then processed for RNA-sequencing and BS-sequencing (Fig 3A). A total of 10 donor mice were analysed (S1 Table). In this assay, we observed significant transcriptome differences between pre and post-graft MuSCs (S3A Fig). Importantly, although muscle regeneration is generally considered to be largely achieved within 28 days post-injury [60], transcriptome analysis showed substantial differences between pre-graft and post-graft TA MuSCs (S3B Fig), indicating that our transplantation of MuSCs *per se* had a direct effect on the gene expression profile of the cells. Genes upregulated after transplantation of TA MuSCs were enriched for immunological processes (S3C Fig), including genes encoding interferon-induced proteins and complement proteins. However, as expected for this time point, the myogenic differentiation marker *Myogenin* was downregulated in the post-graft cells, indicating that cells at the time of re-isolation were not undergoing active myogenic commitment or differentiation. We also observed a slight global increase in the DNA methylation levels after transplantation (mean TA pre-graft: 56.6%, mean TA post-graft: 59.2%; S3D Fig). Whether these observations reflect molecular changes with slow kinetics on MuSCs during muscle regeneration or are the result of the adaptation of engrafted MuSCs to an immunodeficient environment remains to be explored. The latter is favoured as recent RNA-seq analyses of MuSCs during regeneration showed that MuSCs re-acquire a quiescent homeostatic transcriptome as early as 7 days post-injury [61].

To evaluate exclusively the response of EOM MuSCs to a heterologous microenvironment while setting aside modifications in transcriptome due to the transplantation process, we calculated a correction coefficient based on post-graft TA samples and applied it to the

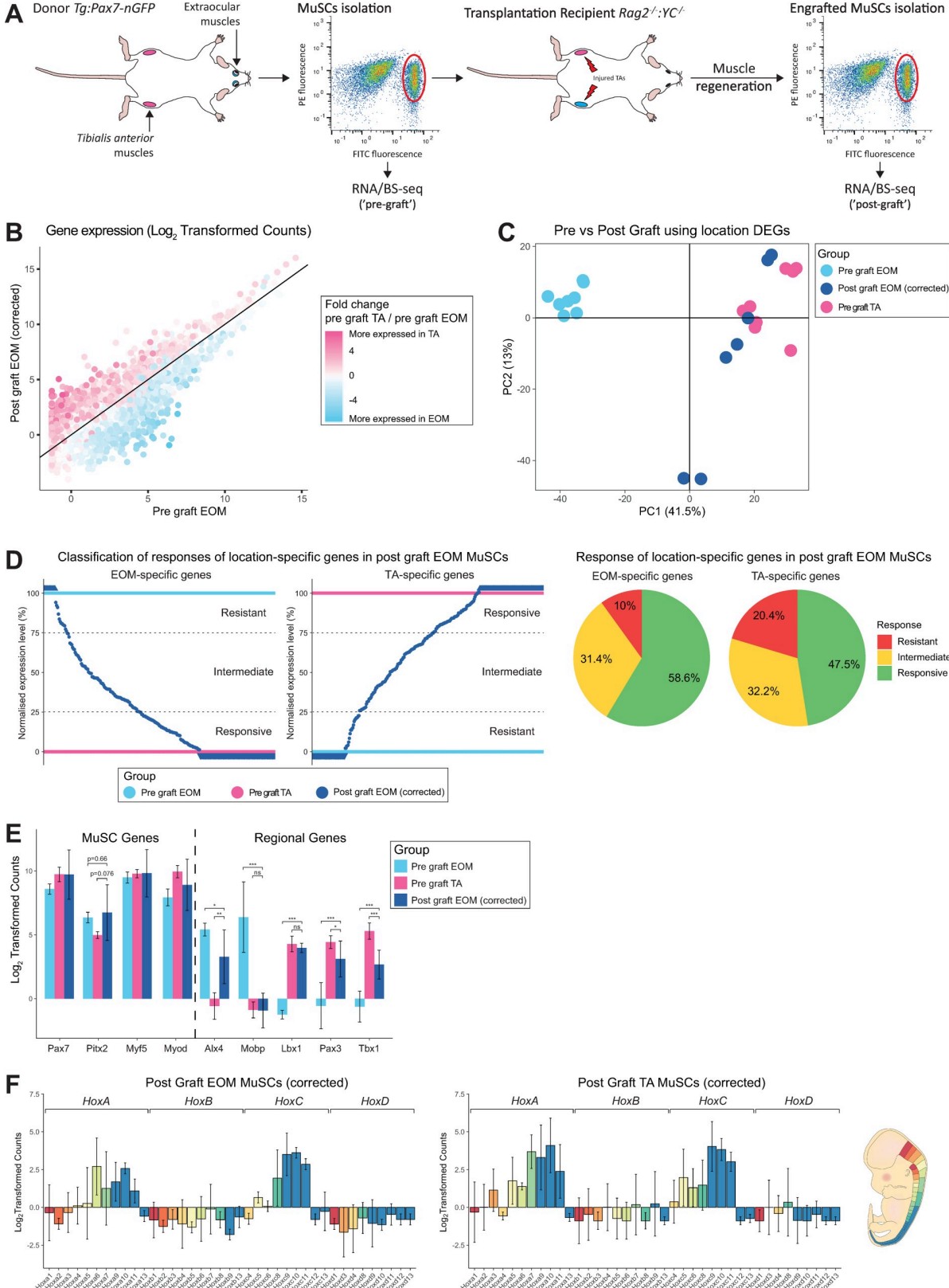

**Fig 3. Head-derived MuSCs adopt a limb-like transcriptome upon transplantation in limb muscles.** (A) Experimental scheme. EOM and TA-derived MuSCs were engrafted into pre-injured recipient TA muscles. After regeneration, engrafted MuSCs were re-isolated and

processed through RNA- and BS-sequencing. 'Post-graft' MuSCs were compared to their 'Pre-graft' counterparts. N = 10 donor mice and N = 10 recipient mice. For each donor mouse, equal number of EOM and TA MuSCs (in the range of 10,000 cells) were transplanted to TA muscles of the same recipient mouse. For each donor muscle, 500 pre-graft cells and 60 re-isolated post-graft cells (mean value) were analysed (S1 Table). (B) Expression analysis of EOM post-graft and EOM pre-graft. Each dot represents a gene. Genes are colour-coded according to their fold change in pre-graft TA *vs* EOM MuSCs (from Fig 1, without transplantation). Note that genes highly expressed in pre-graft TA MuSCs (from Fig 1, coloured in pink) were mostly upregulated in EOM MuSCs following grafting, while genes highly expressed in pre-graft EOM MuSCs (from Fig 1, coloured in blue) were mostly downregulated in EOM MuSCs following grafting. (C) PCA analysis of TA pre-graft, EOM pre-graft and EOM post-graft MuSCs using the expression values of genes differentially expressed between EOM and TA pre-graft MuSCs. PC1 separates EOM and TA MuSCs and shows that after engrafting EOM MuSCs into the TA muscle, they resemble transcriptionally TA MuSCs more than EOM MuSCs. (D) (left) Classification of EOM and TA-specific genes as resistant, responsive, or intermediate upon heterotopic transplantation of EOM MuSCs. Determining the responsiveness of each gene to transplantation was carried out in two steps. First, expression values for TA-specific genes (resp EOM-specific) in pre-graft MuSCs were rescaled to 0–100, where 0 represented the mean expression in EOM (resp TA) MuSCs and 100 represented the mean expression in TA (resp EOM) MuSCs. Next, the expression of EOM and TA-specific genes were rescaled similarly in post-graft EOM MuSCs. Each dark blue dot represents a gene with corresponding rescaled value in post-graft EOM MuSCs. EOM-specific genes with rescaled values in post-graft EOM MuSCs less than 25 were classified as responsive, between 25 and 75 as intermediate and above 75 as resistant. TA-specific genes with rescaled values in post-graft EOM MuSCs less than 25 were classified as resistant, between 25 and 75 as intermediate and above 75 as responsive. (right) Distribution of responses of EOM and TA-specific genes in post-graft EOM MuSCs. (E) Expression of selected markers between TA pre-graft, EOM pre-graft and EOM post-graft MuSCs. Many TA marker genes were upregulated to TA-like levels when EOM MuSCs were grafted into TA muscle. In addition, some EOM marker genes such as *Lmx1a* and *Mobp* were downregulated in this scenario. *** p < 0.001, ** p < 0.01, * p < 0.05 by Welch's t test. (F) Gene expression analysis throughout the *HoxA*, *HoxB*, *HoxC* and *HoxD* clusters in post-graft EOM (heterologous graft, left panel) and TA (homologous graft, middle panel) MuSCs. Genes were colour-coded according to their antero-posterior expression domain in mouse at embryonic day 12.5 (right panel, adapted from [57]). All TA-specific *Hox* genes were upregulated in post-graft EOM MuSCs. See Fig 1E for pre-graft expression levels in EOM and TA MuSCs.

transcriptome of post-graft EOM and post-graft TA samples (see Methods). As expected, the corrected post-graft TA MuSCs clustered with pre-graft TA MuSCs (S3E Fig). Post-graft EOM MuSCs were then corrected similarly and hereafter compared to pre-graft EOM and TA MuSCs (Fig 3B to Fig 3F). We observed a global shift in the transcriptome of post-graft EOM MuSCs towards a TA profile (Fig 3B). The majority of TA-specific genes were upregulated in post-graft EOM MuSCs while EOM-specific genes were downregulated (Fig 3B), indicating that a large proportion of the transcriptome was remodelled by the cellular microenvironment. To further investigate these changes, we performed PCA analysis using location-specific DEGs and observed that post-graft EOM MuSCs tended to cluster closer to pre-graft TA MuSCs than to pre-graft EOM MuSCs (PC1, Fig 3C). The higher similarity of the post-graft EOM samples to pre-graft TA MuSCs was further corroborated by transcriptomic correlation and hierarchical clustering analysis (S3F Fig). The response to grafting EOM MuSCs did show a small amount of heterogeneity, as two samples appeared more centred on PC1 (Fig 3C). This suggests that either the EOM-specific genes were not downregulated sufficiently, or the TA-specific genes were not upregulated sufficiently. In these cases, it is possible that a longer recovery period might have resulted in more a homogeneous outcome. As the quality control metrics of these samples were not different to the remaining samples in this group, this is unlikely to be due to a technical artefact. We then classified location-specific genes in post-graft EOM MuSCs according to their degree of change into three categories: responsive, intermediate, and resistant (Fig 3D left). Genes were classified as resistant if they maintained their initial expression level in EOM MuSCs post-graft, while genes were classified as responsive if they adopted a TA-like expression level in EOM MuSCs post-graft. More specifically, responsive genes included EOM-specific genes fully downregulated to pre-graft TA level or TA-specific genes fully upregulated to pre-graft TA level; intermediate genes were those whose expression levels were between pre-graft EOM and TA samples; and resistant genes included EOM-specific genes that remained at pre-graft EOM level or TA-specific genes that remained at pre-graft EOM level. In agreement with the observed global shift of the transcriptome (Fig 3B), the majority of the location-specific genes showed a responsive or intermediate profile (~90% of the EOM-specific DEGs and ~80% of TA-specific DEGs) while only around 10–20% of DEGs

were resistant after transplantation (Fig 3D right). Importantly, post-graft EOM MuSCs upregulated the TA-specific markers *Pax3*, *Tbx1 and Lbx1* while they down-regulated the EOM-specific markers *Alx4*, *Lmx1a and Mobp* (Fig 3E), confirming their acquisition of a limb-like transcriptional identity. Of note, the EOM-enriched *Pitx2* gene was resistant to alteration in expression following transplantation (Fig 3E). Strikingly, all TA-specific *Hox* genes were upregulated in post-graft EOM MuSCs, with intermediate or responsive behaviours (Fig 3F). This observation is in agreement with previous reports showing that *Hox*-negative tissues or cells adopt the *Hox* status of their new location upon grafting [62,63].

We then followed a similar approach to investigate the response of the epigenome to the different cellular microenvironments in the absence of confounding factors. We first calculated a correction coefficient based on the average DNA methylation differences between the methylomes of pre-graft and post-graft TA MuSCs using windows covering the promoter and enhancer regions and applied it to the methylome of the post-graft MuSCs. The correction coefficient was applied to promoters and enhancers of post-graft EOM and TA MuSCs. After correction, we performed PCA analysis considering promoters of location-specific genes (DEGs). Consistent with our previous results (Fig 2B), we observed no substructure of the data, suggesting no clear changes in DNA methylation at the promoter level (Fig 4A). However, a clear pattern was observed at enhancer regions associated with EOM and TA DEGs. PCA and clustering analysis of these regulatory regions showed that post-graft EOM MuSCs clustered closer to TA samples than to EOM samples (Fig 4B and S4A Fig) indicating that enhancer regions undergo global reprogramming after transplantation in response to the cellular microenvironment. Overall, enhancers associated with EOM DEGs gained DNA methylation whereas those associated with TA DEGs lost DNA methylation in post-graft EOM MuSCs (Fig 4C). These epigenetic changes could contribute to some of the observed gene expression changes upon heterotopic transplantation of EOM MuSCs. Some examples include the EOM DEGs *Eya2* and *Rtn1*, which gain DNA methylation at their associated enhancer and become repressed, and the TA DEGs *Cit* and *Enpep* which lose DNA methylation at their associated enhancer and become more expressed in the post-graft EOM MuSCs (Fig 4D). Interestingly, the TA-specific genes *Lbx1* and *Pax3* showed moderate changes of methylation at their associated enhancers while they became more expressed, indicating that additional mechanisms regulate changes of expression observed in EOM MuSCs upon heterologous transplantation. Of note, changes of enhancer methylation were also observed for some genes showing little variation in expression upon grafting, as for *Myod* and *Pitx2* (S4B Fig). Interestingly, the *HoxA* gene cluster acquired a DNA methylation pattern resembling that of TA MuSCs (Fig 4E), associated with an overall increased expression level (Fig 3F). The other *Hox* clusters did not show much change in their DNA methylation patterns upon heterologous transplantation (S4C Fig). This was not unexpected as EOM and TA MuSCs were found to have similar DNA methylation patterns across these regions. For the *HoxC* cluster, other mechanisms may be responsible for the expression difference observed between EOM and TA MuSCs (Fig 1E) and the gain of expression seen in EOM MuSCs after heterologous transplantation (Fig 3F). For the *HoxB* and *HoxD* clusters, the relatively low methylation levels across these regions may be responsible for the lack of expression in EOM MuSCs, TA MuSCs, and EOM MuSCs after heterologous transplantation (Fig 1E and Fig 3F).

## Discussion

Here we examined the combined transcriptional and DNA methylation signatures of head (EOM) and limb (TA)-derived MuSCs to assess the molecular determinants associated with their diversity and inform on their functional differences such as susceptibility to disease and

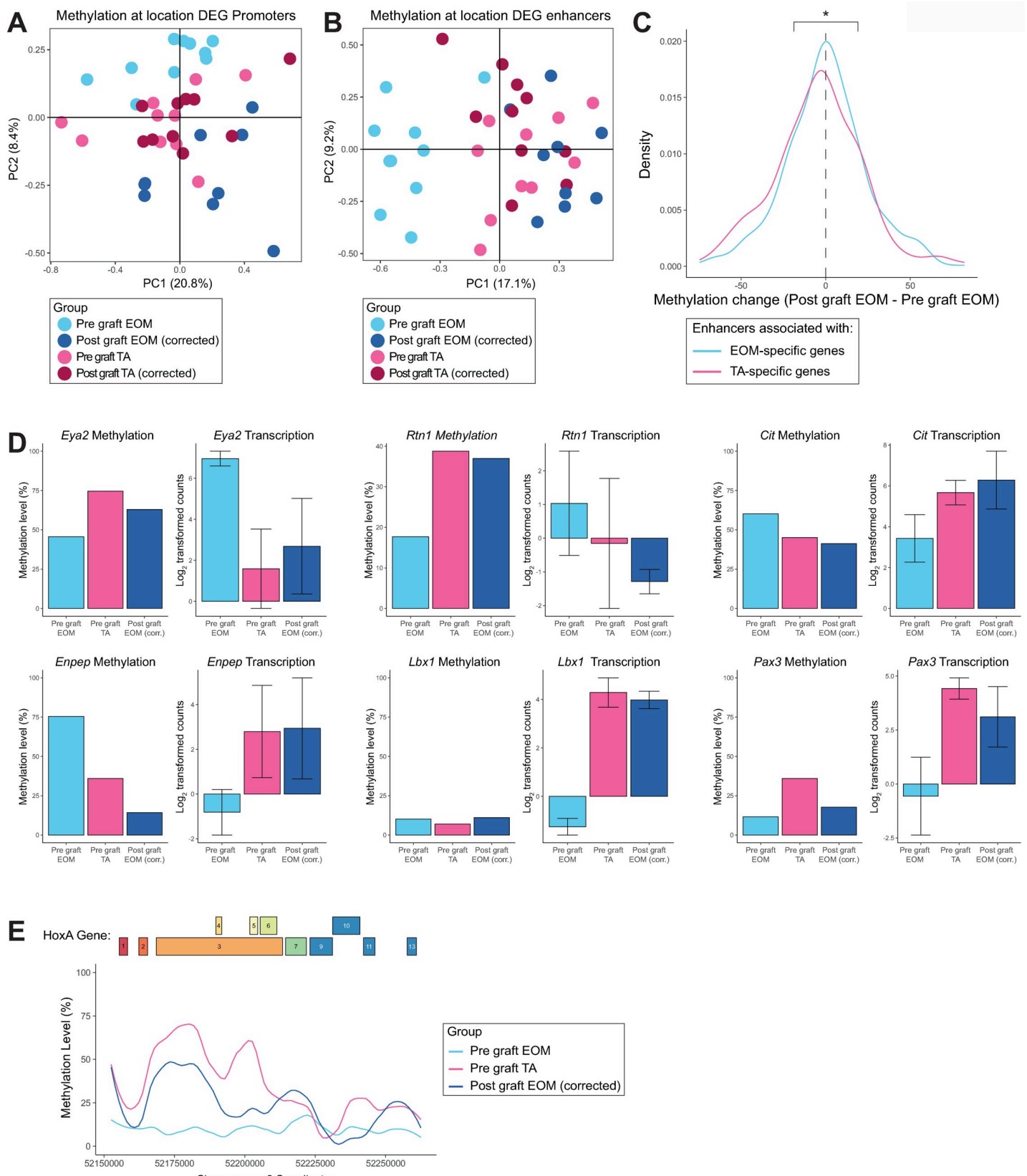

**Fig 4. DNA methylation at enhancers partially accounts for the transcriptome plasticity of MuSCs upon heterotopic transplantation.** (A) PCA analysis of TA pre-graft, EOM pre-graft and EOM post-graft MuSCs DNA methylation at promoters fails to separate samples based on anatomical location. (B) PCA analysis of TA pre-graft, EOM pre-graft and EOM post-graft MuSCs DNA methylation at enhancers separates samples based on anatomical location and demonstrates that EOM MuSCs after grafting resemble TA MuSCs at an epigenetic level in the enhancer context. (C) Density plots of the methylation difference between post-graft EOM MuSCs and pre-graft EOM MuSCs at enhancers associated with EOM or TA upregulated genes. EOM enhancers became hypermethylated in EOM MuSCs after grafting into the TA environment, while TA enhancers were hypomethylated. * p < 0.05 by Student's t test. (D) Enhancer methylation and gene expression levels of selected EOM and TA DEGs that are responsive to grafting. (E) DNA methylation level across the *HoxA* gene cluster in pre-graft EOM, pre-graft TA and post-graft EOM samples. The *HoxA* region was highly methylated in TA MuSCs but not in EOM MuSCs. Notably, DNA methylation was gained across this region when EOM MuSCs were grafted into TA muscle.

regenerative capacity. In this context, we also investigated the extent to which a head-specific identity was determined cell-autonomously through heterotopic transplantation into a limb environment, followed by transcriptome and DNA methylation profiling of re-isolated cells.

Consistent with previous reports, we identified known EOM (*e.g. Alx4*, *Pitx2*) and TA (*e.g. Pax3*, *Lbx1*, *Hox* genes)-specific genes. Surprisingly, we observed expression of *HoxA* and *HoxC* genes, while limb development has been reported to rely mostly on *HoxA* and *HoxD* clusters, with minor contributions from the *HoxB* and *HoxC* clusters [64]. However, we detected expression of *Hoxa11* but not *Hoxa13* in TA MuSCs, consistent with their differential roles in the proximal-to-distal patterning of the limb [64]. Interestingly, we identified *Tbx1* as a TA-specific gene, although it is required for specification of branchial-arch derived craniofacial muscles [30]. This observation might also result from the different fiber-type composition of EOM and TA muscles and the specific properties of myogenic progenitors derived from different muscle fiber-types [65]. Additionally, EOM and TA-specific genes were enriched for developmental processes, which might reflect the persistence of molecular differences acquired during embryogenesis through adulthood. When and to what extent the transcriptomes of head and limb muscle progenitors initially diverge during embryogenesis remains to be addressed.

While several studies investigated the DNA methylation patterns of specific loci (*Myod* [66], *Myogenin* [67], *Desmin* [68], *Six1* [69], *α-smooth muscle actin* [70]) or genome-wide [43–45,71–77] during myogenic specification or myoblast differentiation *in vitro*, we report, to the best of our knowledge, the first genome-wide DNA methylation profiling of purified adult quiescent MuSCs. The adult quiescent MuSCs we analysed here have much lower global methylation levels (~50%) than that reported for somatic cells (~70%) [78], notably muscle fibers (~75%) [79], and other adult stem cells such as intestinal stem cells (78%) [80] and hematopoietic stem cells (84%) [81]. Whether such a low methylation level is required for the stemness and quiescence properties of adult MuSCs remains to be explored, as well as the dynamics of DNA methylation between MuSCs and mature myofibers. Our analysis identified some distinct DNA methylation profiles between EOM and TA MuSCs, notably at enhancers associated with location-specific genes. These results suggest that location-specific transcriptome signatures are determined by location-specific DNA methylation patterns. We are not aware of any previous studies reporting a spatial address of enhancer methylation. It is conceivable therefore that the enhancer epigenome not only registers developmental and tissue-specificity but also the location (along the anterior-posterior axis in this case). Previous reports analysed DNA methylation patterns of similar tissue samples across several anatomical locations [82,83], where changes in cellular composition might be a confounding factor. We report here for the first time the co-analysis of DNA methylome and transcriptome of purified populations of tissue-specific adult stem cells between different anatomical locations.

Our analysis of EOM and TA MuSCs re-isolated after engrafting each population into injured TA muscle followed by regeneration revealed a global reprogramming of EOM transcriptome towards a TA MuSC transcriptome, indicating that the extracellular environment

strongly determines the location-specific signatures we identified. Notably, it has been shown that PW1-positive interstitial cells (PICs) were more abundant in EOM than in TA muscles [84] and that EOM-derived perimysial fibroblasts promoted myotube maturation *in vitro* compared to their limb-derived counterparts [85]. Such differences in the composition of the MuSC niche might explain the specificity and plasticity of transcriptomes we observed. Our finding is in agreement with a previous report [86], indicating that pluripotent stem cell-derived myogenic progenitors remodel partially their molecular signature towards an adult quiescent MuSC transcriptome upon *in vivo* engraftment. Importantly, our data show that TA MuSCs re-isolated after engrafting and regeneration display important transcriptome differences with their pre-grafting TA MuSCs counterparts. This suggests that comparing engrafted cells re-isolated after regeneration to adult quiescent MuSCs [86] could underestimate the extent of transcriptome plasticity driven by the *in vivo* environment.

Significantly, we identified different classes of EOM and TA-specific genes in EOM MuSCs depending on their behaviour following *in vivo* engraftment: resistant, responsive, and intermediate. Genes with intermediate responses could reflect *i)* the stable acquisition of new intermediary expression states due to buffering from initial identity, *ii)* the existence of transcriptional changes with slow kinetics towards full reprogramming or *iii)* the existence of different subpopulations of EOM MuSCs with variable reprogramming efficiency. The first two hypotheses are favored due to the long-term transcriptional changes associated with our experimental procedure and the existence of fully-responsive genes, indicating a complete reprogramming of such genes in each cell.

Transplantation of cranial-derived stem cells allowed us to assess positional information, given that EOM MuSCs lack expression of *Hox* genes. Interestingly, we observed that all TA-specific *Hox* genes were upregulated in EOM engrafted MuSCs, reflecting the acquisition of the *Hox* status of their new location, as observed previously after transplantation of *Hox*-negative cells in a *Hox*-positive domain [63,87] and unlike myoblasts maintaining their axial identity in culture [88], indicating that factors derived from the *in vivo* environment might determine the plasticity of the *Hox* status. Interestingly, MuSCs isolated from the cranial masseter muscle showed a low *Hoxa10* expression compared to TA MuSCs, and this differential expression was maintained two weeks after transplantation into injured TAs [89]. This observation suggests that the acquisition of a new *Hox* status we observed is a slow process, occurring at late stages of muscle regeneration, or that *Hox* genes display different levels of plasticity upon heterologous transplantation of EOM and masseter MuSCs. Whether the initial *Hox*-free status of EOM MuSCs is responsible for their superior regenerative capacity over *Hox*-positive limb-derived MuSCs remains to be determined. The molecular mechanisms of the *Hox* expression and methylation plasticity in adult stem cells remain to be identified, and their manipulation could be of interest in the context of regenerative medicine [90].

We speculate that the limited set of EOM resistant genes could contribute to a niche-independent and cell-intrinsic high engraftment potential of EOM MuSCs. Interestingly, these genes include *Pitx2* and genes involved in thyroid hormone signalling (*Dio2*, encoding the type-II iodothyronine deiodinase, a thyroid hormone activator, and the Tsh hormone receptor *Tshr*). Of note, *Pitx2* overexpression was shown to increase the regenerative capacity of MuSCs [91] and *Pitx2;Pitx3* double mutant mice have impaired muscle regeneration upon injury [92], while thyroid hormone signalling was reported to be critical for MuSC survival and muscle regeneration [93,94]. Therefore, EOM resistant genes could be of interest as deterministic candidate regulators of EOM MuSC identity. Such determinants could be exploited for improving cellular therapy of muscular dystrophies [39,95,96] using the abundant limb MuSCs.

Finally, we identified plastic DNA methylation patterns in engrafted EOM MuSCs, notably at enhancers associated with location-specific genes and at the *HoxA* gene cluster. Importantly, EOM enhancers were annotated based on H3K27ac ChIP data from limb-derived MuSCs [58], which might not fully represent actual enhancers in EOM MuSCs. These observations indicate that cell-extrinsic cues are relayed through specific modifications of DNA methylation patterns to establish a MuSC transcriptome profile matching the new location. Future studies will be required to identify key molecular determinants of this interplay between the niche, the epigenome, and the transcriptome.

In summary, we report the molecular profiling of two populations of adult MuSCs which were reported to exhibit distinct regenerative capacities and originate from muscles that display different disease susceptibility. We identify differences in the methylation of enhancers and demonstrate plasticity upon exposure to a new microenvironment. Therefore, the molecular characterization and functional analysis of MuSCs from multiple muscles and physiopathological conditions will be highly informative for a more complete understanding of muscle stem cell heterogeneity and plasticity.

## Methods

### Ethics statement

Animals were handled according to national and European Community guidelines and an ethics committee of the Institut Pasteur (CETEA, Comité d'Ethique en Expérimentation Animale) approved protocols (Licence 2015–0008).

### Mice

*Tg:Pax7-nGFP* and *Rag2$^{-/-}$;γC$^{-/-}$* mice were used in this study, on C57BL/6;DBA2 F1/JRj and C57BL/6J genetic backgrounds respectively. For experiments, 6 to 8-week-old male littermates were used.

### Isolation of muscle stem cells

Mice were sacrificed by cervical dislocation. *Tibialis anterior* and extraocular muscles were dissected and placed into cold DMEM (ThermoFisher, 31966). Muscles were then manually chopped with scissors and put into a 15 ml Falcon tube containing 10 ml of DMEM, 0.08% collagenase D (Sigma, 11 088 882 001), 0.1% trypsin (ThermoFisher, 15090), 10 μg/ml DNase I (Sigma, 11284932) at 37°C under gentle agitation for 25 min. Digests were allowed to stand for 5 min at room temperature and the supernatants were added to 5 ml of foetal bovine serum (FBS; Gibco) on ice. The digestion was repeated 3 times until complete digestion of the muscle. The supernatants were filtered through a 70-μm cell strainer (Miltenyi, 130-098-462). Cells were spun for 15 min at 515g at 4°C and the pellets were resuspended in 1 ml cold freezing medium (10% DMSO (Sigma, D2438) in FBS) for long term storage in liquid nitrogen or processed directly through FACS-isolation for transplantations.

Before isolation by FACS, samples were thawed in 50 ml of cold DMEM, spun for 15 min at 515g at 4°C. Pellets were resuspended in 300 μl of DMEM 2% FBS 1 μg/mL propidium iodide (Calbiochem, 537060) and filtered through a 40-μm cell strainer (BD Falcon, 352235). Viable muscle stem cells were isolated based on size, granulosity and GFP intensity using a MoFlo Astrios cell sorter (Beckmann Coulter).

Cells were collected in 5 μl cold RLT Plus buffer (Qiagen, 1053393) containing 1U/μl RNAse inhibitor (Ambion, AM2694), flash-frozen on dry ice and stored at -80°C, or in cold DMEM 2% FBS.

## Muscle stem cell transplantations

Muscle injury was done as described previously [97]. Briefly, mice were anesthetized with 0.5% Imalgene/2% Rompun. Both TAs of recipient immunocompromised $Rag2^{-/-};\gamma C^{-/-}$ mice were injured with 50 μl of 10 μM cardiotoxin (Latoxan, L8102) in NaCl 0.9% 24h before transplantation. Muscle stem cells from freshly dissociated EOM and TA muscles of *Tg:Pax7-nGFP* mice were isolated by FACS, spun for 15 min at 515g at 4°C, resuspended in 10 μl of cold PBS and injected into recipient TAs. EOM and TA muscle stem cells from a donor mouse were injected into separate TAs of the same recipient mouse, and equivalent numbers of EOM and TA cells were injected. Transplanted muscle stem cells were re-isolated by FACS based on GFP positivity after three to four weeks.

## Bisulfite-seq and RNA-seq library preparation

RNA was separated from cell lysates and processed into libraries using the G&T method as described in [47]. DNA was also processed into libraries using the bulk protocol described in [48]. Bisulfite-seq libraries were sequenced on an Illumina HiSeq-2500 with 125bp paired end reads and RNA-seq libraries were sequenced on an Illumina HiSeq-2500 with 75bp paired end reads.

## Bisulfite-seq analysis

Adapter sequences and poor-quality calls were removed from the raw sequencing files using Trim galore (version 0.4.2). Reads were then mapped to mouse genome GRCm38 and deduplicated using Bismark (version 0.16.3). Quality control metrics were examined and only samples with a total read count greater than 1000000 reads were taken forward for further analysis.

Promoters were defined as -2000bp to 500bp of the TSS of Ensembl genes. Promoters were split into CpG Island (CGI) promoters and non-CGI promoters based on whether they overlapped a CpG island. H3K27ac peaks were called using macs2 with default parameters on H3K27ac ChIP-seq data obtained from [58]. Enhancers were defined as H3K27ac peaks that did not overlap promoters. Enhancers were linked to genes based on proximity. Enhancers that were further than 1 Mb to the nearest gene were annotated as not linked to any genes. At least two CpG sites needed to be covered by reads for a promoter or enhancer to be taken forward. The mean DNA methylation level was then calculated for each promoter and enhancer that fulfilled this criterion. In addition, the mean DNA methylation of a promoter or enhancer needed to be calculated in at least 3 samples of each experimental group for that promoter or enhancer to be considered in the analysis.

## RNA-seq analysis

Adapter sequences and poor-quality calls were removed from the raw sequencing files using Trim galore (version 0.4.4). Reads were then mapped to mouse genome GRCm38 using Hisat2 (version 2.1.0). Quality control metrics were examined and only samples with a total read count greater than 500000 reads of which at least 70% resided in genes and at least 65% resided in exons were taken forward for further analysis. Read counts were corrected for library size and Log2 transformed using Seqmonk to generate Log2 RPM values. Differential expression analysis was carried out in R using DESeq and genes that demonstrated a fold change greater than 2 and a p-value less than 0.05 were classified as differentially expressed.

Correcting for the effect of transplantation was performed by initially determining the expression change for all genes between TA MuSCs before and after transplantation. This

expression change was then deducted from all post-graft samples to generate corrected values that excluded the effect of transplantation alone.

## Supporting information

**S1 Data. Numerical values for Figs 1–4, S2–S4 Figs.**
(ZIP)

**S1 Fig. FACS profiles of EOM and TA dissociated muscles and gene ontology analysis of the genes upregulated in EOM or TA MuSCs.** (A) FACS profiles of EOM (left) and TA (middle) dissociated muscles. GFP-positive MuSCs (regions outlined in red) were isolated. (right) Overlay of GFP fluorescence intensity of EOM and TA MuSCs. (B) Gene ontology analysis of the genes upregulated in EOM or TA MuSCs. The 10 most significant categories are shown.
(PDF)

**S2 Fig. EOM and TA MuSCs display different DNA methylation patterns at enhancers associated with anatomical location-specific genes.** (A) PCA analysis of DNA methylation of 544 enhancers associated with genes specifically expressed in EOM or TA MuSCs. EOM and TA MuSCs separate according to their location. (B) Scatter plot comparing the mean DNA methylation levels of all enhancers in EOM and TA MuSCs. Enhancers were colour coded as significant if they were found to be differentially methylated by a rolling Z score approach ($p < 0.05$). See Fig 2E for genes with location-specific expression and enhancer methylation.
(PDF)

**S3 Fig. Transplantation of MuSCs induces long-term transcriptome and epigenome modifications.** (A) Whole transcriptome PCA analysis of pre-graft and post-graft TA and EOM MuSCs. Samples separate into before and after grafting. (B) Whole transcriptome PCA analysis of pre-graft and post-graft TA MuSCs only. Pre and post-graft samples still cluster away from each other, indicating that the EOM samples were not solely responsible for the clustering in (A) and that there is a residual impact from the grafting procedure even after the recovery period. (C) GO categories of genes upregulated (left) and downregulated (right) in post-graft *vs* pre-graft TA MuSCs. Upregulated categories suggest there was a remaining inflammatory response. (D) Mean DNA methylation levels in pre-graft and post-graft TA MuSCs for whole genome, repeat elements, promoters and enhancers. Overall there was a global increase in DNA methylation after grafting. (E) Whole transcriptome PCA analysis of pre-graft TA MuSCs and post-graft MuSCs after applying a correction coefficient accounting for transcriptome modifications specifically induced by the transplantation procedure (see Methods). Pre and post-graft samples no longer cluster separately. (F) Hierarchical clustering analysis using Euclidean distance of Pearson correlation values between TA pre-graft, EOM pre-graft and EOM post-graft MuSCs. The samples cluster separately based on anatomical location. Notably, after engrafting EOM MuSCs into TA muscle they cluster with TA MuSCs rather than EOM MuSCs.
(PDF)

**S4 Fig. DNA methylation changes in MuSCs upon heterotopic transplantation.** (A) Hierarchical clustering analysis using Euclidean distance of Pearson correlation values between TA pre-graft, EOM pre-graft and EOM post-graft MuSCs overall clusters samples based on location. Notably, post-graft EOM MuSCs cluster with TA MuSCs rather than with EOM MuSCs. (B) Enhancer methylation and gene expression levels of *Myod* and *Pitx2* prior to and following grafting. (C) DNA methylation level across the *HoxB* (top), *HoxC* (middle) and *HoxD* (bottom) gene clusters in pre-graft EOM, pre-graft TA and post-graft EOM samples. DNA

methylation levels were similar between EOM MuSCs and TA MuSCs. In addition, grafting EOM MuSCs into the TA muscle environment did not substantially affect the DNA methylation across these regions.
(PDF)

**S1 Table. RNA-seq and BS-seq quality control.**
(PDF)

## Acknowledgments

We would like to thank the Flow Cytometry Platform of the Center for Technological Resources and Research (Institut Pasteur) and the Wellcome Trust Sanger Institute sequencing facility for assistance with Illumina sequencing.

## Author Contributions

**Conceptualization:** Brendan Evano, Diljeet Gill, Irene Hernando-Herraez, Thomas M. Stubbs, Wolf Reik, Shahragim Tajbakhsh.

**Data curation:** Diljeet Gill, Irene Hernando-Herraez, Thomas M. Stubbs.

**Formal analysis:** Diljeet Gill, Irene Hernando-Herraez, Thomas M. Stubbs.

**Funding acquisition:** Wolf Reik, Shahragim Tajbakhsh.

**Investigation:** Brendan Evano, Diljeet Gill, Irene Hernando-Herraez, Glenda Comai, Thomas M. Stubbs, Pierre-Henri Commere.

**Methodology:** Brendan Evano, Diljeet Gill, Irene Hernando-Herraez, Thomas M. Stubbs.

**Project administration:** Brendan Evano, Diljeet Gill, Irene Hernando-Herraez, Wolf Reik, Shahragim Tajbakhsh.

**Resources:** Wolf Reik, Shahragim Tajbakhsh.

**Software:** Diljeet Gill, Irene Hernando-Herraez, Thomas M. Stubbs.

**Supervision:** Wolf Reik, Shahragim Tajbakhsh.

**Validation:** Diljeet Gill, Irene Hernando-Herraez, Thomas M. Stubbs.

**Visualization:** Brendan Evano, Diljeet Gill, Irene Hernando-Herraez.

**Writing – original draft:** Brendan Evano, Diljeet Gill, Irene Hernando-Herraez.

**Writing – review & editing:** Brendan Evano, Diljeet Gill, Irene Hernando-Herraez, Glenda Comai, Wolf Reik, Shahragim Tajbakhsh.

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
