## [Decision Letter · Decision Letter 0]

10 Jul 2020

Dear Dr Tajbakhsh,

Thank you very much for submitting your Research Article entitled 'Transcriptome and epigenome diversity and plasticity of muscle stem cells following transplantation' to PLOS Genetics. Your manuscript was fully evaluated at the editorial level and by independent peer reviewers. The reviewers appreciated the attention to an important topic but identified some aspects of the manuscript that should be improved.

We therefore ask you to modify the manuscript according to the review recommendations before we can consider your manuscript for acceptance. Your revisions should address the specific points made by each reviewer.

[LINK]

Yours sincerely,

John M. Greally, D.Med., Ph.D.

Section Editor: Epigenetics

PLOS Genetics

Gregory Barsh

Editor-in-Chief

PLOS Genetics

Reviewer's Responses to Questions

**Comments to the Authors: **

Reviewer #1: The manuscript by Evano et al. describes transcriptome and methylome differences between muscle stem cells from mouse hindlimb (TA) and extraocular muscles (EOMs) before and after transplantation into hindlimb muscles. Previous work from this group showed EOM muscle stem cells successfully engrafted into injured hindlimb muscle, but, based on expression of a small number of EOM-specific genes, failed to maintain the EOM muscle phenotype in their new location. They have followed this work up in the current paper by generating transcriptomes and DNA cytosine methylomes from TA and EOM muscle stem cells, pre- and post-engraftment into the TA. Consistent with their earlier work, EOM stem cells were reprogrammed after engraftment to a transcriptome resembling TA stem cells. A small percentage of EOM-specific genes were resistant to reprogramming, but the majority were highly responsive or of an intermediate character. The very interesting conclusion is that the local muscle microenvironment has a major role in dictating specific characteristics of stem cells associated with a given muscle, at a given body location.

The findings in this paper are straightforward and convincing. While it would be great if the reverse experiment could be performed (i.e., TA stem cells engrafted into the eye), this seems extremely difficult, if not impossible, with mice. My additional comments are minor.

1.It would be worth speculating on what potential differences between the TA and EOM environment could play a role in reprogramming during regeneration.

2.In Figure 3, two of the corrected post-graft EOM muscle samples are outliers, sitting on the 0 point for PC1. Can the authors comment on this?

3.On line 266, page 10, it is stated, “… indicating that stem cell-derived myogenic progenitors…”. It would avoid confusion to rephrase this as “… indicating that pluripotent stem cell-derived myogenic progenitors…”.

Reviewer #2: In this manuscript, Evano et al. address the significant question of the differences between muscle satellite cells derived from distinct lineages (limb vs extraocular muscle) across multiple axes. In particular, the authors explore the molecular determinants of functional differences between limb and EOM satellite cells by identifying transcriptional and epigenetic signatures that differ between them and determining the extent to which these signatures are cell-intrinsic or determined by their surrounding niche. 

Data presented here are novel additions to knowledge in the field, particularly the whole-genome epigenetic profiling of quiescent satellite cells. In addition, the discovery that gene expression and methylation patterns can be largely redetermined by niche factors in cross-transplanted cells suggests an intriguing degree of plasticity. 

The figures are nicely clear and convey the experimental results logically. Some comments:

Figure 2A- As presented, the differential methylation data are somewhat underwhelming. 

Figure 2E- While it is not unexpected in experiments of this sort, it would be nice if the authors would address genes (such as Adamtsl1) characterized by both significant over- and under-methylation when comparing limb and EOM-derived cells.

Figure 3F- It would assist in interpretation if this panel also included data from pre-grafted TA and EOM satellite cells. If the data are already in-hand, inclusion of results from homologously grafted cells (TA at least) would also improve the impact of the figure/interpretation.

The main concern I have with the writing is the perhaps excessive speculation/overinterpretation of some aspects of the results in the Discussion section, although this may be a matter of personal taste. However, in the context of this interesting if speculative discussion the lack of follow up for any of the genes identified and discussed as potential key determinants of the phenotype is somewhat of a missed opportunity.

**Have all data underlying the figures and results presented in the manuscript been provided?** Large-scale datasets should be made available via a public repository as described in the *PLOS Genetics*
data availability policy, and numerical data that underlies graphs or summary statistics should be provided in spreadsheet form as supporting information.

Reviewer #1: **No: **The authors state that theta will be deposited with GEO after acceptance.

Reviewer #2: Yes

PLOS authors have the option to publish the peer review history of their article (what does this mean?). If published, this will include your full peer review and any attached files.

Reviewer #1: No

Reviewer #2: No

---

## [Decision Letter · Decision Letter 1]

2 Aug 2020

Dear Dr Tajbakhsh,

We are pleased to inform you that your manuscript entitled "Transcriptome and epigenome diversity and plasticity of muscle stem cells following transplantation" has been editorially accepted for publication in PLOS Genetics. Congratulations!

Yours sincerely,

John M. Greally, D.Med., Ph.D.

Section Editor: Epigenetics

PLOS Genetics

Gregory Barsh

Editor-in-Chief

PLOS Genetics

Comments from the reviewers (if applicable):

Reviewer's Responses to Questions

**Comments to the Authors:**

Reviewer #1: The authors have addressed my comments in a satisfactory manner.

Reviewer #2: The authors have addressed all of my comments.

**Have all data underlying the figures and results presented in the manuscript been provided?**

Reviewer #1: Yes

Reviewer #2: Yes

PLOS authors have the option to publish the peer review history of their article (what does this mean?). If published, this will include your full peer review and any attached files.

Reviewer #1: No

Reviewer #2: No

**Data Deposition**

http://datadryad.org/submit?journalID=pgenetics&manu=PGENETICS-D-20-00788R1

**Press Queries**

---

## [Editor Report · Acceptance letter]

19 Oct 2020

PGENETICS-D-20-00788R1 

Transcriptome and epigenome diversity and plasticity of muscle stem cells following transplantation 

Dear Dr Tajbakhsh, 

We are pleased to inform you that your manuscript entitled "Transcriptome and epigenome diversity and plasticity of muscle stem cells following transplantation" has been formally accepted for publication in PLOS Genetics! Your manuscript is now with our production department and you will be notified of the publication date in due course.

With kind regards,

Matt Lyles

PLOS Genetics

On behalf of:
